# Postprandial Hypotension and Spinal Cord Injury

**DOI:** 10.3390/jcm10071417

**Published:** 2021-04-01

**Authors:** Rikke Middelhede Hansen, Klaus Krogh, Joan Sundby, Andrei Krassioukov, Ellen Merete Hagen

**Affiliations:** 1Spinal Cord Injury Centre of Western Denmark, Department of Neurology, Regional Hospital, DK-8800 Viborg, Denmark; joan.sundby@midt.rm.dk; 2Department of Hepatology and Gastroenterology, Aarhus University Hospital, DK-8200 Aarhus, Denmark; klaukrog@rm.dk; 3University of British Columbia and GF Strong Rehabilitation Centre, Department of Medicine, International Collaboration on Repair Discovery, Vancouver, BC V5Z 1MP, Canada; krassioukov@icord.org; 4National Hospital for Neurology and Neurosurgery, Queens Square, UCLH, London WC 1N 3BG, UK; ellenmerete.hagen@nhs.net; 5Institute of Neurology, University College London, London WC 1N 3BG, UK

**Keywords:** spinal cord injury, postprandial hypotension, food ingestion, ambulatory blood pressure measurement, cohort study

## Abstract

Postprandial hypotension (PPH) is defined as a fall of ≥20 mmHg in systolic blood pressure (SBP) or a SBP of <90 mmHg after having been >100 mmHg before the meal within two hours after a meal. The prevalence of PPH among persons with spinal cord injury (SCI) is unknown. Ambulatory blood pressure measurement was performed in 158 persons with SCI, 109 men, median age was 59.1 years (min.:13.2; max.: 86.2). In total, 78 persons (49.4%) had PPH after 114 out of 449 meals (25.4%). The median change in SBP during PPH was −28 mmHg (min.: −87; max.: −15 mmHg) and 96% of the PPH episodes were asymptomatic. The occurrence of PPH was correlated to older age (*p* = 0.001), level of injury (*p* = 0.023), and complete SCI (*p* = 0.000), but not, gender or time since injury. Further studies are needed to elucidate if PPH contributes to the increased cardiovascular mortality in the SCI population.

## 1. Introduction

In most individuals, the intake of a meal increases blood flow to the gut with no or very minor effects on systemic blood pressure. Abnormally low systolic blood pressure (SBP) following a meal is termed postprandial hypotension (PPH). Usually PPH is defined as a fall of ≥20 mmHg in SBP or a SBP of <90 mmHg after having been >100 mmHg before the meal within two hours after ingesting a meal [1,2,3]. The prevalence of PPH increases with age and may occur as a side-effect to various medications [4,5,6]. In addition, PPH is associated with increased risk of falls, syncope, coronary events, stroke, asymptomatic lacunar infarction, asymptomatic cerebrovascular damage, and death [7]. The above mentioned are associated with a risk of vascular cognitive impairment and dementia [8].

Spinal cord injury (SCI) has profound effects on autonomic function, including disruption of baroreflex and cardiovascular regulation [9,10,11]. Loss of supraspinal control may cause increased blood pressure (BP) variability, orthostatic hypotension (OH), exercise-induced hypotension, post-exercise induced hypotension, autonomic dysreflexia, and reduced quality of life [12,13,14,15,16,17,18,19,20,21]. Ambulatory blood pressure measurement (ABPM) has been used in several studies of fluctuations in the BP in various groups of patients, including persons with SCI [22,23,24]. Case reports have indicated that PPH occurs in persons with SCI, but the prevalence is unknown and potential associations with level and type of SCI remain uncertain [25,26,27,28,29].

In the western world, the incidence of non-traumatic SCI and the age at time of injury have increased significantly [30,31,32]. Fortunately, the expected longevity of persons with SCI has also increased dramatically over the last five decades. Today, cardiovascular disease, urinary tract infection, and septicemia are the leading causes of increased mortality in persons surviving acute SCI [33]. It is plausible that BP instability and PPH may contribute to the increased cardiovascular mortality and increased risk of dementia seen in the SCI population [34].

Spinal cord injury severely affects gastrointestinal function, causing delayed gastric emptying and colonic transit time [35,36,37,38,39]. Splanchnic blood flow is under autonomic control, but it remains unknown how SCI affects postprandial blood flow to the intestines.

The primary aim of the present study was to describe the prevalence of PPH in a large group of persons with SCI by means of ABPM. The secondary aim was to determine whether PPH is associated with age, gender, time since SCI, or the level and completeness of lesions.

## 2. Materials and Methods

### 2.1. Subjects

The present study is a cohort study based on ABPM performed among persons with SCI admitted to The Spinal Cord Injury Centre of Western Denmark. The center covers the western half of Denmark with an underlying population of 3.1 million inhabitants and receives persons with both non-traumatic and traumatic SCI in all age groups for specialized rehabilitation after the acute phase. ABPM is part of the systematic routine assessment of hospitalized persons with a new SCI. It is also used as part of outpatient follow-up when persons with SCI report symptoms of autonomic dysfunction. Subjects included in the present paper were investigated from January 2017 to May 2017 or from October 2019 until restrictions due to the COVID-19 pandemic were instituted in March 2020. From medical records, data were obtained regarding date of birth, gender, date of injury and ABPM, type of injury (traumatic/non-traumatic), neurological level of injury, and ASIA Impairment Scale (AIS) grade according to International Standards of Neurological Classification of Spinal Cord Injuries [40]. Information of gastrointestinal morbidity and surgery was obtained from the medical records or the International Spinal Cord Injury Bowel Function Basic Data Set (version 2.0) when available [41]. Data on diabetes and other neurological diseases (i.e., Parkinson’s disease, multiple system atrophy, and stroke) as well as number of medications used on the day of the ABPM were obtained for each patient.

Use of data for publication was granted from the Hospital Management, Regional Hospital Central Jutland. The project was reported to The Scientific Ethical Committees of Region Central Jutland (case number 1-10-72-181-20) and The Legal Office of the Central Denmark Region (reference number 706735, case number 1-16-02-590-20).

### 2.2. Ambulatory Blood Pressure Measurements

The ABPM was recorded using Meditech Card(X)plore device and Cardiovision 1.1.8.22 software (Meditech Ltd., Budapest, Hungary). The sampling frequency was four times per hour during the daytime (07:00–22:00) and two times per hour during the night. Figure 1 shows an example of an ABPM with three episodes of PPH in a 69-year old male with a C 1, AIS B SCI four months and eight days post injury. The person was one of seven persons having three or four episodes of PPH during the ABPM.

Each person with SCI was instructed to fill in a diary of activities, including time of meals and symptoms of hypo- and hypertension i.e., dizziness, headache, feeling weak, sweating, and blurred vision. Patients unable to fill in the diary were assisted by the staff. At the centre, three main meals are served at 08:00 (breakfast), 12:00 (lunch), and 17:30 (dinner). These time points were used as time of assumed meal intake, unless comments from the diary confirmed meals at another time. If eating was noted in the diary, this was considered the correct time of a meal. The following activities were noted in the diary as well: physical activity e.g., physical exercise, physiotherapy and occupational therapy, transfer (transfers with and without lift), eating at other times than the main meals, and nutrition through percutaneous endoscopic gastrostomy-tube (PEG-tube). Information of clean intermittent catheterization, bowel management, dressing/personal care, reposition in bed, smoking, administration of medication, drinking at other times than at meals, fluid through PEG-tube, procedures related to tracheostomy, continuous positive airway pressure (CPAP)/Bilevel Positive Airway Pressure (BiPAP), and resting were obtained as well, but not used in the present study.

### 2.3. Statistical Analysis

The applied definition of PPH was a decrease in SBP ≥20 mmHg or SBP <90 mmHg if SBP before the meal was ≥100 mm Hg within two hours after ingesting a meal [1,2,3]. The mean value of SBP measurements one hour before a meal and with at least two measurements was used as the reference SBP. Logged SBP data, including diary notes of activities, were transferred to Excel, and coded into the defined activities. Data from Excel were augmented with a calculated SBP drop indicator for each time-stamped SBP measurement using a Python script and in addition enriched with demographic and medication data prior to importing the data in STATA via Stat Transfer 14. STATA version 16 was used for preprocessing the data and statistical analysis. Various Python scripts were used on csv exported data to generate figures and extract counting statistics for physical activity and transfer events before and after meals. Logistic regression was performed using PPH as the dependent variable. Independent variables were gender, age at ABPM, number of meals, number of medications on the day of ABPM, time since injury, complete/incomplete SCI, gastrointestinal morbidity and surgery, other neurological diseases, diabetes and level of injury defined as: (1) high tetraplegia with neurological level of injury from C1–C3, (2) low tetraplegia C4–C8, (3) high paraplegia (T1–T6), and (4) low paraplegia (T7 and below).

## 3. Results

Valid recordings of pre- and postprandial SBP were available for a total of 449 meals in 158 subjects. Data regarding demography, comorbidity, and use of medications are shown in Table 1. The median observational time of the ABPMs was 24 h. The median SBP was 120 mmHg before breakfast, 125 mmHg before lunch, 125 mmHg before dinner, and 118 mmHg during the night.

A total of 114 (25.3%) episodes of decrease in SBP within two hours after a meal meet the criteria of PPH in 78 (49.4%) subjects. In only seven (4.4%) subjects, the decrease in SBP was associated with symptoms of hypotension. The median time from ingestion of the meal until PPH was registered was 60 min (min 15, max 120 min). The median change in SBP during PPH was −28 mmHg (min: −87; max: −15 mmHg). Twenty of 114 (17%) episodes interpreted as PPH occurred simultaneously with transfers noted in the diary, while 26 (23%) occurred simultaneously with physical activity e.g., physical exercise, physiotherapy, and occupational therapy. Logistic regression analysis revealed that PPH was associated with age when ABPM was performed, higher levels of injury, and complete SCI (Table 2).

## 4. Discussion

The main finding from the present study is that PPH is common among persons with SCI. In our setting, the prevalence of PPH was 49% and the risk increased with increasing age, higher levels of SCI, and completeness of the lesion. Surprisingly, concomitant medication, other neurological diseases or diabetes did not increase the risk. To the best of our knowledge, the present study is the first to provide data on PPH in a large group of persons with SCI.

Our results are in contrast to some earlier publications. Catz et al. found that PPH can occur in thoracic paraplegia, but not in tetraplegia [29]. Baliga et al. did not find PPH in neither para- nor tetraplegic patients [28]. The present study is by far the largest but also differs from previous by having a high proportion of incomplete and non-traumatic lesions and especially by including significantly older patients. The previous studies were performed under standardized conditions with liquid test-meals and supine position during study and with a post prandial observational time of 45 min. In contrast, our data were collected during the patient’s daily routine and most recordings lasted 24 h. In support of our findings, two case reports on persons with SCI and age 62 and 66 years also found PPH [26,27]. It is therefore likely that age is an important factor for developing PPH in persons with SCI.

The cardiovascular response to a meal is mediated through the sympathetic gastrovascular reflex, whereby gastric distension elicits a vasoconstrictive response. In addition, the plasma levels of glucose, insulin, and norepinephrine rises. In supine young able-bodied subjects, ingestion of a meal leads to minor, if any change, in SBP. The increased splanchnic blood flow in the superior mesenteric artery is counterbalanced by increased heartrate, cardiac output, and systemic peripheral resistance. In supine elderly subjects, ingestion of a meal leads to the same changes, but the SBP decreases if the subject is sitting.

The pathophysiology of PPH is not fully understood but it represents an imbalance between increased splanchnic blood flow and the needed adjustment from the cardiovascular system. Impaired peripheral vasoconstrictor response, lack of increase in heartrate by activation of the sympathetic nervous system i.e., reduced baroreflex function and diminished function of the gastrovascular reflex are factors known to contribute. The temperature of the meal is important, as fluid with a temperature of 50 °C affects the SBP more than a fluid meal of 5 °C [42]. PPH is more frequent after breakfast than lunch and dinner due to circadian changes in BP [43]. Furthermore, a high content of mono-saccharides, primarily glucose, and a high number of calories transported to the duodenum also increases risk of PPH [44].

In other groups of patients, PPH is often asymptomatic [3]. Despite the lack of symptoms, PPH is associated with an increased risk of falls, syncope, coronary events, stroke, asymptomatic lacunar infarction, asymptomatic cerebrovascular damage, and death [4,7,45]. In our study, PPH was observed in 49% of patients, but only 4% had symptoms. Worldwide, people with SCI are living longer and the risk of cardiovascular diseases is high [46]. Furthermore, persons with SCI are at an increased risk of developing non-Alzheimer’s dementia [34]. Thus, PPH may be clinically important, even if asymptomatic. PPH is defined as occurring within two hours after a meal [1]. Our study shows that a drop in SBP can occur from 15–120 min after ingesting a meal. We have not examined if a decrease in SBP could last more than 120 min after a meal. Studies have found the most pronounced fall in SBP after 60 min [47,48]. These studies show that the lowering of BP post meal continues, to a lesser extent, after two hours in persons with autonomic failure and essential hypertension. Persons with SCI have delayed gastric emptying [35]. This could raise the question if the observation time post meal should be longer than two hours. However, the findings from the present study need to be confirmed and clinically relevant interventions developed before changes in daily practice of persons with SCI and PPH are recommended. Thus, the definition of PPH must be validated for the SCI-population. This includes definitions for a standard test-meal, test position, sampling intervals, length of observational time after ingesting a meal, time of day for the test, and which medication should be paused. Possibly, this could be performed as an addition to the International Standards to Document Remaining Autonomic Function after Spinal Cord Injury [49]. Long-term observational studies are needed to elucidate if PPH is associated with increased morbidity and mortality in persons with SCI. Further investigations are needed to explore the mechanisms that trigger PPH, specifically in the SCI population.

There are several limitations to the present study. We performed ABPM for approximately 24 h. Among patients in hospital, we assumed that they actually ate their meals at the time it was served. In addition, we do not know the exact composition of each meal and the amount of liquid ingested. At our institution, meals are served at specific time points and patients were encouraged to write in the diary, if they deviated from this daily routine. Since we defined PPH as a decrease in SBP occurring within 2 h after a meal and most episodes occurred within this time frame, we find that the exact timing of the meal is of lesser importance. However, the fall in SBP in the postprandial observational time could arise from other activities known to trigger hypotensive episodes i.e., transfers and physical activity. In our study, such activity could potentially explain up to 50% of episodes interpreted as PPH. In spite of this, our data indicate that PPH is very common in persons with SCI. Another limitation is the medication taken by the subjects during the study period. Thus, some cases of PPH may be explained by medication taken, rather that SCI per se. Taken as a whole, the use of medication was not associated with PPH, but we did not go into details with each type of drug or combination of medications.

In conclusion, we found that PPH is common among persons with SCI. We also found that PPH is associated with higher levels of SCI, complete lesions, and age of the patient.

## Figures and Tables

**Figure 1 jcm-10-01417-f001:**
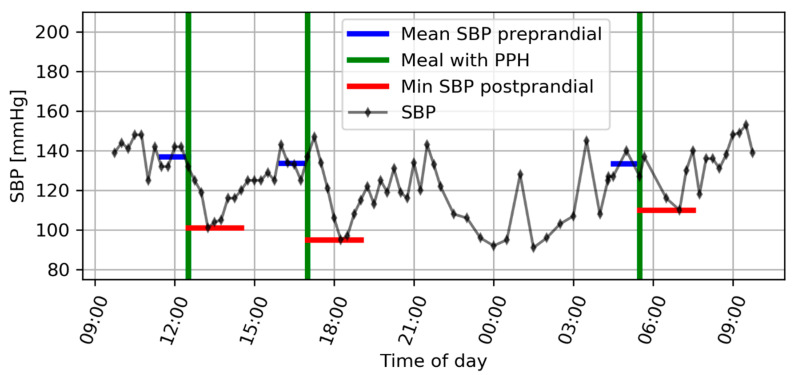
ABPM showing systolic blood pressure (SBP) in a 69-year old male with C1, AIS B SCI four months and eight days post injury. He had three episodes of PPH after lunch, dinner, and early breakfast.

**Table 1 jcm-10-01417-t001:** Demographics, comorbidity, and medications when ABPM was performed.

*N*= 158	Number (*n*)	Frequency (%)
Male/Female	109/49	69%/31%
Tetraplegia		
High/low tetraplegia ^1^	49/45	31%/28%
AIS A	14	9%
AIS B	8	5%
AIS C	13	8%
AIS D	59	37%
Paraplegia		
High/low paraplegia ^1^	26/38	16%/24%
AIS A	15	9%
AIS B	3	2%
AIS C	12	8%
AIS D	34	22%
Non-traumatic SCI	85	54 %
Diabetes	16	10%
Other neurological diseases	20	13%
Number of medications	Median: 10	(min: 0; max: 22)
Time since injury	Median: 0.24 years	(min: 0.02; max: 56.7)
Age	Median: 59.1 years	(min: 13.2; max: 86.2)
Duration of ABPM	Median: 24.0 h	(min: 10.5; max: 49.4)

^1^ High Tetraplegia C1–C3, low tetraplegia C4–C8, high paraplegia T1–T6, low paraplegia T7 and below. ABPM: Ambulatory blood pressure measurement. SCI: Spinal Cord Injury. AIS: ASIA Impairment Scale.

**Table 2 jcm-10-01417-t002:** Logistic regression analysis.

Independent Variable	Odds Ratio	Standard Error	*p*-Value	95% Confidence Interval
Age when ABPM performed	1.039	0.012	0.001 *	1.015–1.063
Gender	0.950	0.434	0.906	0.404–2.231
Time since spinal cord injury	1.068	0.466	0.132	0.980–1.163
Level of injury ^1^	1.512	0.120	0.023 *	1.060–2.160
Complete/incomplete SCI	9.482	5.941	0.000 *	2.776–32.380

Statistical method: Logistic regression with PPH as the dependent variable. Number of medications on the day of the ABPM, gastrointestinal morbidity and surgery, other neurological diseases and diabetes were used as independent variable as well but did not show statistically significant association with PPH. ^1^ Level of injury was divided into four groups: high tetraplegia C1–C3, low tetraplegia C4–C8, high paraplegia T1–T6, low paraplegia T7 and below. * *p*-values < 0.01 are considered statistically significant.

## Data Availability

The data presented in this study are available on request from the corresponding author. The data are not publicly available due to language.

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
