# Peer review of "Postprandial Hypotension and Spinal Cord Injury"

_jcm, 2021, doi:10.3390/jcm10071417_

Round 1

Reviewer 1 Report

The study titled “Postprandial hypotension and spinal cord injury” examines the prevalence of postprandial hypotension (PPH) by means of ambulatory blood pressure measurement in a cohort of 158 individuals having a spinal cord injury.  Associations of PPH with age, gender, time since injury and level/completeness are also examined. The study is well designed. The results are clearly presented and provide very important information on the cardiovascular systems response to a meal. There are just a few minor issues and grammatical errors that need to be addressed prior to publication.

Methods Section 2.2, line 88: “Figure 1 shows an example….” Is this a “typical” example or is the one presented showing an example from those having a maximal effect? Please specify.

Figure 1 Legend, line 89: “…in a subject…”. Please specify age, gender, AIS Grade and years post-injury for the research participant whose data is shown.

Discussion line 185: “Worldwide, the age of the SCI-population increases…” Unclear what this means. Are you trying to say that people with SCI are living longer? Please edit.

Line 33: extra space between “ingesting” and “a meal”

Line 72: there should be a comma after “records”

Line 79: “on the day” instead of “at the day”

Lines 91,96,97,110: “diary” instead of “dairy”

Line 98: activity – lower case

Line 191: “nadir”??

Reviewer 2 Report

This is a very interesting research paper about postprandial hypotension after spinal cord injury (SCI). The authors recorded the pre and postprandial SBP for 449 meals in 158 patients. From the data collected, the authors concluded that postprandial hypotension is associated with age and the severity of SCI. These findings are interesting and can be used in clinics to serve the patients with SCI better.

Revision

In the Results section, the authors reported the logistic regression analysis but didn’t show any graph, it will be much better to make graphs over the results.

Since the study is a cohort study not a case-control study, to report the relative risk based on the data collected is more suitable. I am wondering whether the authors can calculate the relative risks and make graphs over the results as well.

Reviewer 3 Report

The manuscript is a clinical study, in which postprandial hypotension was investigated in SCI patients. Authors reported that the occurrence of PPH was correlated to age and complete SCI but not to level of injury, gender or time since injury. In the discussion, authors explained possible mechanisms underlying this symptom. The experiment was appropriately designed, and results were supported by strong data evidence. This manuscript adds novel insights to the field of SCI.

Only one minor error: an extra space in line 33 on the 1st page.
